# Religious Fundamentalism, Delusions, and Conspiracy Beliefs Related to the COVID-19 Pandemic

**DOI:** 10.3390/ijerph19159597

**Published:** 2022-08-04

**Authors:** Małgorzata Sobol, Marcin Zajenkowski, Konrad S. Jankowski

**Affiliations:** Department of Psychology, University of Warsaw, 00-183 Warsaw, Poland

**Keywords:** religious fundamentalism, delusions, conspiracy beliefs, COVID-19

## Abstract

The widespread COVID-19 conspiracy theories are a problem in dealing with the pandemic, as their proponents tend not to adhere to public health regulations. The aim of this study was to analyse the relationships between religious fundamentalism, delusions, compliance with public health regulations, and religion-related conspiracy beliefs about the COVID-19 pandemic. The participants were 570 internet users aged 18–60. They responded to questions regarding sociodemographic variables, compliance with public health regulations, conspiracy beliefs concerning COVID-19, as well as the Revised Religious Fundamentalism Scale, and the Delusions Scale. The results indicated that people exhibiting more conspiracy beliefs were less likely to comply with public health regulations concerning the COVID-19 pandemic and showed more religious fundamentalism. Additionally, there was an indirect effect of religious fundamentalism on conspiracy beliefs through delusions. The results suggest that when formulating epidemiological messages, it is worth paying attention to the importance of rational thinking.

## 1. Introduction

One of the most important issues in the fight against the COVID-19 pandemic is compliance with the principles of preventing the spread of the disease [1,2,3]. However, conspiracy beliefs regarding COVID-19 present a problem [4]. Research shows that such beliefs are associated with a disregard for restrictions [4,5]. Conspiracy theories about COVID-19 concern the origins of the coronavirus and the belief that the pandemic situation is being used by those in power to manipulate people or the economy [4,5].

Conspiracy beliefs are sometimes associated with religious content [6]. The religious aspect might be even more pronounced in the COVID-19 conspiracy theories because of the church attendance restrictions. Indeed, COVID-19 conspiracy beliefs have been linked to religious fundamentalism [7]. It often happens that in places of worship, epidemiological recommendations are not followed as it is believed that in the church it is not possible to become infected [8]. There are situations in which the police must intervene because the faithful flock to their temples despite epidemiological prohibitions [9]. Such religious attitudes are associated especially with a specific form of religiosity, religious fundamentalism, which is understood as a belief that the rules of one’s religion are irrefutable and should therefore be given priority in the hierarchy of values [10]. Previous studies have shown connections between Christian nationalism and ignoring epidemiological recommendations [8]. Research conducted in the United States showed that supporters of liberal parties more often respected the restrictions related to the COVID-19 pandemic, while supporters of right-wing religious groups ignored these restrictions more often than supporters of other groups [8,11,12]. Indeed, stay-at-home orders were less adhered to in more religious communities in the United States during the COVID-19 pandemic [13], and attendance at religious services was related to both more COVID-19 infections and more deaths [14].

Religious beliefs in extremely strong and rigid forms might have a lot in common with delusions. Delusions are understood as rationally unjustified representations and interpretations of events and behaviour of others, and one’s own thoughts [15]⁠. Delusions are an important diagnostic criterion for mental disorders, but they are experienced to a low degree by mentally healthy people [16]. Religion and delusions connect, inter alia, beliefs that are not based on a verifiable intersubjective reality [17]. In the case of religion, the beliefs are shared by part of society; in the case of delusions, beliefs are specific to a given person. The research results indicate significant positive relationships between delusions and some forms of religiosity [17], and a positive correlation between the tendency to delusions and the intensity of conspiracy beliefs [4,18].

The aim of the current study was to analyse associations of religious fundamentalism [10] and delusions [15] with compliance with public health regulations and the religion-related COVID-19 conspiracy beliefs. Based on theoretical considerations and the research presented above, the following hypotheses have been formulated:

**Hypothesis** **1** **(H1).**
*Religion-related COVID-19 conspiracy beliefs are negatively correlated with compliance with public health regulations concerning the COVID-19 pandemic.*


Previous studies [4,5] revealed that beliefs that the pandemic situation is being used by those in power to manipulate people or the economy are related to disregard for restrictions related to the COVID-19 pandemic. We aimed to test if similar associations are present in the case of the COVID-19 conspiracy beliefs associated with religious content.

**Hypothesis** **2** **(H2).**
*Delusional thinking mediates the association between religious fundamentalism and religion-related conspiracy beliefs about COVID-19.*


The second hypothesis concerns relationships between religious fundamentalism, delusional thinking, and the COVID-19 conspiracy beliefs associated with religious content. Religious fundamentalism requires that everyday behaviour be instilled with the belief that religious principles are right and irrefutable. Such an attitude requires not listening to rational thoughts and weakening the sensitivity to one’s own feelings and emotions related to the current situation. Strong religious beliefs that are not subjected to critical thinking can generalize to other spheres of thought. That is, a person with rigid religious beliefs becomes more inclined to adopt irrational views about various spheres of life. To maintain fundamentalist religious beliefs, critical, rational thinking must be turned off. Then the mind is more prone to adopting irrational beliefs from various spheres, i.e., it is prone to delusional thinking. Therefore, we expected that religious fundamentalism entails delusional thinking that is detached from reality. In turn, people with a tendency to delusional thinking are more prone to adopting various types of conspiracy theories based on irrational beliefs with a high emotional charge.

The abovementioned mechanisms might be of particular importance in countries with high religiosity rates, because in such countries more COVID-19 infections are observed, even after considering the control of a country’s gross domestic product per capita [14]. Poland is an example of a developed country with 85% of the population identifying as Roman Catholic [19]. Roman Catholicism in Poland is largely Marian and revelatory with numerous services based on revelations and ascribing magical sacred power to places, objects, and rituals [20].

## 2. Method

### 2.1. Participants

There were 570 participants (290 females, 280 males), aged 18–60 years (M = 37.70; SD = 11.80), all of whom were Polish. Most participants had secondary school education (56.3%); others had university (31.4%) or primary school education (12.3%). Respondents lived in big cities (12.6%), big towns (18.1%), medium towns (20.4%), small towns (13.2%), or villages (35.8%). Participants were registered in the nationally representative online panel ARIADNA. Respondents were randomly chosen from this database. The surveyed group was a representative sample of Poles using the Internet. Participation in the research was rewarded with points in the panel’s loyalty program. The study protocol received approval from the research ethics committee. Data were collected in November 2020. The CAWI method (computer-assisted web interview) was used to ask 140 questions in total.

### 2.2. Measures

The Revised Religious Fundamentalism Scale [10], was used to measure religious fundamentalism, defined as the belief that only one set of religious teaching contains the truth about deity and humanity, and that people who follow this truth have special relations with the deity. It consisted of 12 items (e.g., ‘To lead the best, most meaningful life, one must belong to the one, fundamentally true religion’), with a 9-point Likert type scale (from −4 ‘strongly disagree’ to 4 ‘strongly agree’).

The Delusions Inventory (PDI [15] in the Polish adaptation [18] was used to measure delusional ideation in the nonclinical population. The questions aim to examine a lifetime experience. The PDI consists of 40 items (e.g., ‘Do you ever feel as if other people can read your mind?’). Respondents rate whether beliefs are present or absent. There are the following categories: delusions of control; misinterpretations, delusions of reference; delusions of persecution; expansive delusions; delusions concerning various types of influence; other delusions; delusions based on guilt, depersonalisation, and hypochondria; thought reading and broadcast.

Compliance with the public health regulations was measured using nine questions with a three-point Likert-type response format [21]. These questions concerned hygiene and social distance (see Appendix A) as implied by the recommendations of the Polish government to prevent COVID-19 spread.

Conspiracy beliefs concerning COVID-19 were measured using six questions created for the purpose of this study. These questions concerned beliefs related to the conspiracy of religious forces. Specifically, we asked participants whether they agree (1 = ‘not at all’ to 100 = ‘very much’) with statements that: 1. the pandemic was an action of Satan to distract people from the Church; 2. the church is immune to coronavirus; 3. the pandemic is fabricated to distract people from church service; 4. the greatest harm the coronavirus causes is the destruction of religious life; 5. a person who truly believes in God does not adhere to the restrictions and fear over the coronavirus; 6. the pandemic is a punishment sent by God. The items were highly correlated (average *r* = 0.64; *p* < 0.001) and summarised to create an index of religious conspiracy beliefs.

## 3. Results

In Table 1 we present correlations and descriptive statistics. We found that conspiracy beliefs were correlated with lower levels of compliance with public health regulations, and with higher levels of fundamentalism and delusions. Fundamentalism correlated positively with delusions and compliance with public health regulations.

Subsequently, we tested the hypothesis that delusions would mediate the relationship between religious fundamentalism and the endorsement of COVID-19 conspiracy beliefs related to religious reality. We controlled for age and sex. We used the PROCESS macro [22] with 5000 bootstrapped samples and a 95% confidence interval for the indirect effect of religious fundamentalism. We standardised the scores prior to analysis. The results revealed that religious fundamentalism was positively associated with delusions (b = 0.15, SE = 0.04, *p* < 0.001, 95% CI [0.07, 0.23]). Delusions were positively associated with conspiracy (b = 0.19, SE = 0.04, *p* < 0.001, 95% CI [0.12, 0.26]). Once delusions were entered into the model, the direct effect of religious fundamentalism was still significant (b = 0.43, SE = 0.03, CI [0.36, 0.51]). The indirect effect was also significant (b = 0.03, SE = 0.01, CI [0.01, 0.05]). Thus, delusions partially mediated the relationship between fundamentalism and conspiracy beliefs.

## 4. Discussion

The aim of the current study was to test the relationships between religious fundamentalism, delusions, compliance with public health regulations, and the COVID-19 conspiracy beliefs, associated with religious content. The findings supported the first hypothesis. Religious conspiracy was negatively correlated with compliance with public health regulations concerning COVID-19. The hypothesis concerning the indirect effect of religious fundamentalism on conspiracy beliefs through delusions was also supported. Thus, people with high religious fundamentalism are more likely to endorse conspiracy theories, in the domain which is most important to them (i.e., the spiritual realm). The stronger the religious fundamentalism of a person, the more often he or she has delusional thoughts, i.e., thoughts that are irrational or detached from reality, usually with a high emotional charge [15]. Such thinking, in turn, promotes the endorsement of conspiratorial beliefs about COVID-19, which are irrational and highly emotional [4,5].

Among the important factors predisposing people to endorse conspiracy theories, researchers typically name perceived powerlessness [23], lack of control [24], and uncertainty [25]. Believing in irrational explanations of the causes of events gives people the feeling of control and satisfies their need to see the world around them as predictable and organised [24]. Religious beliefs might serve this function as well. Religion, especially in its fundamental form, gives answers to basic existential questions and provides the rules by which one should live in order to have a sense of peace and security [10]. Beliefs in God are associated with perceiving the world as a place where everything is planned and controlled [26]. Thus, religious people might seek to explain random and threatening events as a consequence of secret forces. Indeed, conspiracy beliefs are positively associated with religiosity [6]. The need for answers to difficult existential questions and the feeling of peace and security are especially important in life-threatening situations [27], such as the COVID-19 pandemic [28]. Therefore, presumably, people with strong fundamental religiosity with a tendency for delusional thinking adopt conspiratorial beliefs about COVID-19 because they give them specific answers to questions about the causes of the current situation and its further course, thus reducing anxiety and the sense of threat.

The current study is not free of limitations. First, the single-measurement character of the study should be emphasised. A diary-based design would be useful in further studies. Second, our sample consisted of Polish participants only, thus, care should be taken in generalising the results to other cultures. Another limitation of the research is that other religiosity-related variables, such as attendance and private devotional activities, were not controlled for.

## 5. Conclusions

In conclusion, the results of this study indicate that people with higher (vs. lower) levels of religious fundamentalism were more likely to have delusions and, consequently, showed stronger conspiracy beliefs associated with religious content about COVID-19. The results of our study can be used in practice. Psychological work on rational thinking could be used to prevent the development of conspiratorial beliefs in highly religious people as these religion-related conspiracy beliefs may contribute to disregard of epidemiological recommendations to prevent the spread of COVID-19.

## Figures and Tables

**Table 1 ijerph-19-09597-t001:** Correlations and Descriptive Statistics of All Variables.

	1	2	3	*M*	*SD*	α
1. Fundamentalism	-			48.46	18.30	0.91
2. Delusions	0.15 *** (0.16 ***)	-		49.50	9.68	0.95
3. Compliance with the regulations	0.10 * (0.10 *)	−0.01 (0.03)	-	69.40	23.95	0.90
4. Religious conspiracy	0.45 *** (0.47 ***)	0.28 *** (0.26 ***)	−0.14 *** (−0.09 *)	126.23	107.92	0.91

Note: *** *p* < 0.001, * *p* < 0.05. Partial correlations controlling for age and sex are presented in parentheses.

## Data Availability

Data are available at https://osf.io/2mqw9/ (accessed on 16 July 2022).

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
