# Peer review of "Religious Fundamentalism, Delusions, and Conspiracy Beliefs Related to the COVID-19 Pandemic"

_ijerph, 2022, doi:10.3390/ijerph19159597_

Round 1

Reviewer 1 Report

This manuscript deals with an interesting topic, the relationships between religious factors and COVID-19 conspiracy theories, focusing on a sample of 570 Polish adults. Although the topic is important, I encountered several difficulties and concerns while reviewing this paper:

I still do not fully understand the connection between religious fundamentalism and delusions. Do the authors mean to say that these are two types of "irrational" thinking? If the authors are developing a causal model, then it would be helpful to know more about how or whether fundamentalism gives rise to delusional ideation.

It would be useful to include more information (besides citation of another work) when describing the key measures. The brief descriptions provided were inadequate in my view.

It seemed to have Table 1 displayed, but no Table 2 containing hypothesized model(s), or relevant multivariable models, etc. There was no information given about any statistical controls. Given the potential for confounding, did the authors control for education or other social or demographic characteristics? To be sure the observed patterns really reflect the distinctive role of fundamentalism only, it would be useful to control for other facets of religiosity (e.g., attendance, private devotional activities, etc.).

I am not expert on the Polish context but in the U.S. (where a good deal of this research on religion-COVID links has been generated), it is not at all clear whether belief structure (i.e. fundamentalism) is more important in giving rise to religious conspiracy theories. Instead, it may be specific theological strands and organizational forms work together to spread and support religious conspiracy theories. In particular, independent charismatic and fundamentalist churches with autonomous and powerful leaders (preachers) seem to traffic extensively in religious conspiracy theories about COVID.

The authors might be interested in several studies over the past 12 months or so by Terrence Hill and his associates on religion and COVID-related protective lifestyles, which use novel data to elucidate particular pathways.

Author Response

This manuscript deals with an interesting topic, the relationships between religious factors and COVID-19 conspiracy theories, focusing on a sample of 570 Polish adults. Although the topic is important, I encountered several difficulties and concerns while reviewing this paper:

  1. I still do not fully understand the connection between religious fundamentalism and delusions. Do the authors mean to say that these are two types of "irrational" thinking? If the authors are developing a causal model, then it would be helpful to know more about how or whether fundamentalism gives rise to delusional ideation.

Response 1. We expanded our explanation of the relationship between religious fundamentalism and delusional thinking.

  1. It would be useful to include more information (besides citation of another work) when describing the key measures. The brief descriptions provided were inadequate in my view.

Response 2. We added more information about measures.

  1. It seemed to have Table 1 displayed, but no Table 2 containing hypothesized model(s), or relevant multivariable models, etc. There was no information given about any statistical controls. Given the potential for confounding, did the authors control for education or other social or demographic characteristics?

Response 3. We agree that additional demographic variables should be controlled for in the analyses. The literature on religiosity suggests that especially sex and age might be important for religious beliefs (e.g., Bengston et al., 2015; Miller & Hoffmann,1995). Thus, in Table 1, we added partial correlations between the study variables controlling for age and sex (the coefficients in parentheses). Moreover, we controlled for age and sex in the mediation model. The results did not change much – the direction and significance of the associations remained the same.

References

Bengston, V. et al. (2015). Does Religiousness Increase with Age? Age Changes and Generational Differences Over 35 Years. Journal for the Scientific Study of Religion, 54, 363-379.

Miller, A. S., & Hoffmann, J. P. (1995). Risk and religion: An explanation of gender differences in religiosity. Journal for the Scientific Study of Religion, 34(1), 63–75.

  1. To be sure the observed patterns really reflect the distinctive role of fundamentalism only, it would be useful to control for other facets of religiosity (e.g., attendance, private devotional activities, etc.).

Response 4. We agree it would be best to have these measures to get more confidence in the results, but we did not measure these variables, therefore we added one more limitation in the discussion section: “Another limitation of the research is that other religiosity-related variables such as attendance and private devotional activities were not controlled.”

  1. I am not expert on the Polish context but in the U.S. (where a good deal of this research on religion-COVID links has been generated), it is not at all clear whether belief structure (i.e. fundamentalism) is more important in giving rise to religious conspiracy theories. Instead, it may be specific theological strands and organizational forms work together to spread and support religious conspiracy theories. In particular, independent charismatic and fundamentalist churches with autonomous and powerful leaders (preachers) seem to traffic extensively in religious conspiracy theories about COVID.

Response 5. In our model of relations between variables, we indeed can see reasons for the susceptibility to the adoption of conspiracy theories, in the irrationality and rigidity of thinking characteristic of religious fundamentalism. The research was conducted among adherents of one religion - Catholicism. You are right, because, in a sense, Polish local Catholic churches can be described as an independent charismatic and fundamentalist church with autonomous and powerful leaders. We added this note to the Discussion section.

  1. The authors might be interested in several studies over the past 12 months or so by Terrence Hill and his associates on religion and COVID-related protective lifestyles, which use novel data to elucidate particular pathways.

Response 6. Thank you for indication. We referred to the work of Terrence Hill in the Discussion section.

Reviewer 2 Report

Thank you for submitting this paper. These are my suggestions

1) please briefly describe in the abstract the main methods applied (right now there is only the nr of respondents) = "online survey" or similar should be added

2) Line 32-33: is there a source (newspaper article/academic literature) for this affirmation?

3) Method: it could be better described, as a non polish person I am not familiar with some things, e.g. what is panel ARIADNA? What is the "polish internet user population"? Please consider adding a source or a footnote. I suggest first of all to describe the sample: where were participants recruited, how were they interviewed? The online survey (I presume) had to be filled in autonomously or with an interviewer? If there is a source please add it. How many questions did the survey have in total? Moreover, why was Poland chosen as a case study? Maybe a few words on the case study (levels of religiosity and so on) would be useful to contextualise the research.

4) Measures: "compliance with covid-9 regulations" is referred to an online appendix I don't have access to, but indeed it would be useful to have the full survey that was used in the appendix, so that the study could be replicated in other contexts.

5) In the conclusion please highlight possible practical implications of the findings e.g. for policy makers or religious institutions (there is a hint of that in the abstract, so I was expecting it in the review and indeed it makes sense to have it).

The fact that beliefs in COVID-19 conspiracy theories negatively predict responsible pandemic-related behaviour is not surprising and there is a lot of literature on that. Bringing into the picture the impact of religious fundamentalism is indeed an interesting perspective, however also not fully new. Please consider reading and referring e.g. to this paper: https://www.sciencedirect.com/science/article/pii/S0191886921007923

The overall review is interesting, however after reading it there are curiosities that are left. E.g. is there a correlation between the results and some demographic information? E.g. gender, age, place of residence (rural VS urban). This might be missing because this work is not a full article, but rather a short review. However, it would be interesting to know if these variables yielded any impact on the results or whether this will be investigated in the next step of the research.

Author Response

Thank you for submitting this paper. These are my suggestions

1) please briefly describe in the abstract the main methods applied (right now there is only the nr of respondents) = "online survey" or similar should be added

Response 1. We added more information about the main methods applied.

2) Line 32-33: is there a source (newspaper article/academic literature) for this affirmation?

Response 2. We added the source for this affirmation.

3) Method: it could be better described, as a non polish person I am not familiar with some things, e.g. what is panel ARIADNA? What is the "polish internet user population"? Please consider adding a source or a footnote.

Response 3. We added the link to the panel Ariadna. We explained more clearly what the “polish internet user population” means.

3.1) I suggest first of all to describe the sample: where were participants recruited, how were they interviewed? The online survey (I presume) had to be filled in autonomously or with an interviewer? If there is a source please add it.

Response 3.1. We added more information about the sample.

3.2) How many questions did the survey have in total?

Response 3.2. We added the following sentence: “The survey had 140 questions in total.”

3.3) Moreover, why was Poland chosen as a case study? Maybe a few words on the case study (levels of religiosity and so on) would be useful to contextualise the research.

Response 3.3. We explained why Poland was chosen as a case study.

4) Measures: "compliance with covid-9 regulations" is referred to an online appendix I don't have access to, but indeed it would be useful to have the full survey that was used in the appendix, so that the study could be replicated in other contexts.

Response 4. Full survey is in the appendix.

5) In the conclusion please highlight possible practical implications of the findings e.g. for policy makers or religious institutions (there is a hint of that in the abstract, so I was expecting it in the review and indeed it makes sense to have it).

Response 5. We highlighted possible practical implications of the findings.

5.1) The fact that beliefs in COVID-19 conspiracy theories negatively predict responsible pandemic-related behaviour is not surprising and there is a lot of literature on that. Bringing into the picture the impact of religious fundamentalism is indeed an interesting perspective, however also not fully new. Please consider reading and referring e.g. to this paper: https://www.sciencedirect.com/science/article/pii/S0191886921007923

Response 5.1) We referred to this paper.

5.2) The overall review is interesting, however after reading it there are curiosities that are left. E.g. is there a correlation between the results and some demographic information? E.g. gender, age, place of residence (rural VS urban). This might be missing because this work is not a full article, but rather a short review. However, it would be interesting to know if these variables yielded any impact on the results or whether this will be investigated in the next step of the research.

Response 5.2) Thanks for this suggestion. We have included age and sex as control variables in Table 1 (partial correlations in parentheses) as well as in the mediation model. The results remained similar (see also our response to comment of Reviewer 1).

Round 2

Reviewer 1 Report

no additional comments at this time